# Constrained Probabilistic Diffusion Model for Seismic Data Reconstruction Using a Restoration Operator Based on a Deep Image Prior

## Abstract

Seismic data acquisition is of the utmost importance in the oil industry, as it allows the representation of subsurface geological features. Several factors affect the number of sources or receivers during the acquisition, impacting seismic data quality. Different methodologies have been developed for reconstruction based on generative models for seismic signals. Methods based on diffusion models (DM) have emerged in this context for seismic data reconstruction, guiding the image generation or the reverse process to solve the reconstruction problem using a closed-form solution and deep learning-based solvers. However, a disadvantage of these methodologies is that they cannot extract all the features necessary to represent the data domain, which is crucial for accurate reconstruction. As a result, the entire DM must be re-trained for each experiment, leading to high computational costs due to its complexity. We propose DM-RODIP, an alternative DM approach for seismic data reconstruction, where the reverse process of a pre-trained DM is guided toward a reconstruction problem solution using a restoration operator based on a Deep Image Prior. The proposed method was evaluated on synthetic and field data, demonstrating superior reconstruction performance with improvements of up to 10.2 dB in PSNR and 0.09 in SSIM for synthetic data, and 1.0 dB in PSNR and 0.04 in SSIM for field data, outperforming state-of-the-art methods.

## 1 Introduction

Seismic data plays a crucial role in the oil industry, enabling the representation of subsurface geological characteristics through wave propagation generated by sources. These waves, reflected in two (2D) or three dimensions (3D), measure the behavior of the ground along a surface, providing information for oil and hydrocarbon exploration, such as characterizing reservoirs, geological faults, and fluid detection (Hatton et al., 1986; Evans & Dragoset, 1997). The seismic acquisition involves distributing a series of sources across the area of interest, generating waves that propagate through the subsurface. The energy reflected or transmitted by the subsurface layers is captured by a set of receivers or geophones (Hatton et al., 1986; Herrera et al., 2010; Evans & Dragoset, 1997; Chaouch & Mari, 2006). The coordinates of the sources and receivers, arranged with uniform and dense spacing, define the designed geometry (pre-plot) to ensure high-quality seismic exploration. This directly impacts seismic processing and the generation of high-quality subsurface images, which are of great interest to the hydrocarbon industry.

However, during acquisition, the geometry may vary (post-plot) due to operational, economic, geological, or environmental factors in seismic surveys, which can affect the number of sources or receivers and potentially limit seismic acquisition (Hatton et al., 1986; Herrera et al., 2010). As there is a possibility of losing information or needing to add more information to the dataset than planned in the pre-plot geometry to achieve higher resolution, there is an image restoration problem seen from the computational perspective; therefore, various methodologies have been proposed for the reconstruction of seismic information sampled both regularly and irregularly (Li et al., 2017; Liu et al., 2022).

Different interpolation or estimation techniques have been developed for these reconstructions based on filtering methodologies, wavefield operators through transforms (Fourier), and sparsity algorithms (Liu et al., 2015; Yang et al., 2012; Kazemi et al., 2016). Deep learning-based algorithms have emerged as powerful tools in many applications due to their ability to learn increasingly complex representations of information from large volumes of data. Currently, a variety of methods in the field of deep learning (DL) have been applied to seismic data reconstruction using different architectures for computational learning models (Chai et al., 2020; Mandelli et al., 2018). These architectures have proven to be a handy tool for feature extraction and better solutions for the image restoration problem compared to conventional methods (Kelleher, 2019). In particular, Deep Image Prior (DIP) addresses seismic reconstruction when sampling is irregular, traces are missing, and matched training data are scarce or geologically mismatched; it fits an untrained convolutional network directly to the observed data, letting the architecture act as an implicit prior that favors coherent seismic events and suppresses incoherent noise (Ulyanov et al., 2018; Rodríguez-López et al., 2023; Liu et al., 2021).

In the realm of DL, different methodologies have been proposed for solving inverse problems based on generative models (Asim et al., 2020; Dimakis, 2022; Shah & Hegde, 2018; Goyes-Peñafiel et al., 2024), as these models, once trained, capture a large number of low-level features and information suitable for providing a solution. In this context, diffusion models (DM) are a technique used to generate images based on learning a distribution of the data domain (Ho et al., 2020), and state-of-the-art research has demonstrated the ability of these models to solve inverse problems (Chung et al., 2022b; Song et al., 2023; Chung et al., 2022a; Song et al., 2022). Different techniques attack the seismic data reconstruction problem in two ways: finding the solution of the inverse problem during diffusion model training (Wang et al., 2024a; Gong et al., 2025), or conventionally training the diffusion model to generate images and guiding the reverse process (in the inference or generation step) to a solution (Zhu et al., 2023). From this last approach, methodologies have been proposed for the reconstruction of seismic data, with the disadvantage that the proposed algorithms cannot extract all the features necessary to represent the entire data domain to subsequently perform the restoration task. Therefore, they require retraining the whole model for each experiment, which generates a high computational cost due to the complexity of these models (Deng et al., 2024; Wang et al., 2024b; Durall et al., 2023; Wei et al., 2024; Wang et al., 2025; Goyes-Peñafiel et al., 2025).

Our method (DM-RODIP) is inspired by Goyes-Peñafiel et al. (2025), but introduces key differences. Specifically, we propose a more robust reconstruction framework by guiding the diffusion model with a restoration operator that leverages both classical restoration principles and deep learning. This allows for improved reconstruction quality and better generalization across different seismic scenarios. The proposed method was evaluated using synthetic and field data, demonstrating superior reconstruction performance compared to state-of-the-art methods.

## 2 BACKGROUND

### 2.1 DEEP RESTORATION PRIORS

Deep Restoration Priors (DRP) generalize denoiser-based priors by using any pre-trained restoration network $\mathcal{M}_{\phi^*}$ as an implicit prior with optimized parameters $\phi^*$, enabling state-of-the-art restoration models (e.g., super-resolution) to regularize different inverse problems within a principled framework. The method integrates the measurement model $\mathbf{y} = \mathbf{H}\mathbf{x} + \sigma^2$ with a plug-and-play-like iteration driven by $\mathcal{M}_{\phi^*}$, where the unknown signal $\mathbf{x} \in \mathbb{R}^m$ is corrupted by a measurement operator $\mathbf{H} \in \mathbb{R}^{m \times n}$ to obtain the measurements $\mathbf{y} \in \mathbb{R}^n$ with a Gaussian noise $\sigma^2$ and $m > n$; providing a convergence analysis to stationary points of a global functional associated with the implicit prior (Hu et al., 2023).

DRP iterates $\mathbf{z}_k = \mathbf{x}_{k-1} - \gamma\tau(\mathbf{x}_{k-1} - \mathcal{M}_{\phi}(\mathbf{H}\mathbf{x}_{k-1}))$, to condition the solution with the prediction of $\mathcal{M}_{\phi^*}$, which is then followed by a scaled proximal operator:

$$\mathbf{x}_k = \text{prox}_{\gamma f}^{\mathbf{H}^{\mathsf{T}}\mathbf{H}}(\mathbf{z}_k) = \text{argmin}_{\mathbf{x}} \frac{1}{2} \|\mathbf{x}_{k-1} - \mathbf{z}_k\|_2^2 + \frac{\gamma}{2}\|\mathbf{y} - \mathbf{H}_{\mathbf{D}}\mathbf{w}_t\|_2^2, \tag{1}$$

where the second term represents the data fidelity term, ensuring measurement consistency and yielding an efficient solution (Kamilov et al., 2023).

## 2.2 DEEP IMAGE PRIOR

Deep Image Prior (DIP) demonstrates that the architecture of an untrained convolutional generator already encodes sufficient information to capture a large amount of low-rank image statistics, yielding promising results in solving standard inverse problems (e.g., super-resolution and inpainting) (Ulyanov et al., 2018). By applying this methodology, only the observed data are required to train the network. This is the main advantage of this approach, which leads to a lower computational cost and allows faster and more efficient reconstruction (Ulyanov et al., 2018; Rodríguez-López et al., 2023; Liu et al., 2021).

DIP replaces explicit regularization by reparameterizing $\mathbf{X} = \mathcal{M}_\phi(\mathbf{Z})$ and solving $\phi^* = \operatorname{argmin}_\phi \|\mathbf{H}(\mathcal{M}_\phi(\mathbf{Z})) - \mathbf{Y}\|_2^2$, with a fixed random input $\mathbf{Z}$ and output $\hat{\mathbf{X}} = \mathcal{M}_{\phi^*}(\mathbf{Z})$ optimized by gradient descent from random $\phi$, where $\mathbf{H}$ simulates the degradation depending on the specific task.

## 2.3 DIFFUSION MODELS

A DM consists of two main components: the forward and reverse processes. The main goal of this approach is to model the distribution of the data and generate realistic images from noise. In the forward process, Gaussian noise is gradually added to an image over multiple steps until it becomes pure noise. In the reverse process, a DL-based model learns to remove the noise at each step, guiding the reconstruction of the noisy image back to its original state or generating a synthetic one with a distribution similar to the original data.

### 2.3.1 FORWARD PROCESS

In the forward process, noise following a Gaussian probability distribution is gradually introduced to the image in multiple steps to transform a clean data point $\mathbf{X}_0$ into a latent variable $\mathbf{X}_T \sim \mathcal{N}(0, 1)$. This process is defined by a Markov chain $q(\mathbf{X}_t|\mathbf{X}_{t-1})$ represented by the normal distribution $\mathcal{N}(\mathbf{X}_t; \sqrt{\alpha_t}\mathbf{X}_{t-1}, \beta_t\mathbf{I})$, where $\beta_t \in (0, 1)$ is a variance schedule to control the noise added at each step $t$, $\alpha_t = 1 - \beta_t$ quantifies the clean data fraction of $\mathbf{X}_{t-1}$ that remains in $\mathbf{X}_t$ after a diffusion step, and the total number of steps is denoted by $T$, in this way $t \in \{0, 1, ...T\}$. With the reparameterization trick and the transition probability, $\mathbf{X}_t$ can be directly estimated from $\mathbf{X}_0$, assuming $\mathbf{X}_t \sim q(\mathbf{X}_t|\mathbf{X}_0)$, as follows,

$$\mathbf{X}_t = \sqrt{\bar{\alpha}_t}\mathbf{X}_0 + \sqrt{1 - \bar{\alpha}_t}\epsilon_t, \quad \boldsymbol{\epsilon}_t \sim \mathcal{N}(0, 1), \tag{2}$$

where $\bar{\alpha}_t = \prod_{s=0}^t \alpha_s$ represents the fraction of the clean data point $\mathbf{X}_0$ that remains after $t$ diffusion steps.

### 2.3.2 REVERSE PROCESS

In the reverse process, a neural network $\boldsymbol{\epsilon}_\theta(\mathbf{X}_t, t)$ learns the noise $\boldsymbol{\epsilon}$ for each step $t$ to return from $\mathbf{X}_T$ to $\mathbf{X}_0$. A Markov chain $p_\theta(\mathbf{X}_{t-1}|\mathbf{X}_t)$ represents this process and follows the normal distribution $\mathcal{N}(\mathbf{X}_{t-1}; \mu_\theta(\mathbf{X}_t, t), \sqrt{\beta_t})$, where $\mu_\theta(\mathbf{X}_t, t)$ is derived from the network noise predictions and $\sqrt{\beta_t}$ from the forward noise schedule.

Finally, the reverse process requires both the $\boldsymbol{\epsilon}_\theta(\mathbf{X}_t, t)$ network and the forward noise schedule to compute $\mathbf{X}_{t-1}$ from $\mathbf{X}_t$ following the normal distribution equation

$$\mathbf{X}_{t-1} = \frac{1}{\sqrt{\alpha_t}}\left(\mathbf{X}_t - \frac{\beta_t}{\sqrt{1 - \bar{\alpha}_t}}\boldsymbol{\epsilon}_\theta(\mathbf{X}_t, t)\right) + \sqrt{\beta_t}\boldsymbol{\epsilon}. \tag{3}$$

### 2.3.3 DM TRAINING AND TESTING

To train the DM, the reparameterization trick allows the objective to be expressed to minimize the mean squared error (MSE) between the Gaussian noise $\epsilon_t$ controlled by the variance schedule $\beta_t$, and the noise predicted by the network $\epsilon_\theta(\mathbf{X}_t, t)$ for each step $t$. The loss function can be derived in a simplified form as shown in the following equation according to Ho et al. (2020):

$$\theta^* = \operatorname*{argmin}_\theta \mathbb{E}_{\mathbf{X}_0, \boldsymbol{\epsilon}_t, t}\|\boldsymbol{\epsilon}_t - \boldsymbol{\epsilon}_\theta(\mathbf{X}_t, t)\|^2. \tag{4}$$

To generate new images, only the network $\boldsymbol{\epsilon}_{\theta*}(\mathbf{X}_t, t)$ is required, such that we obtain $\mathbf{X}_0$ from $\mathbf{X}_T$. To achieve this, $\mathbf{X}_{t-1}$ must be computed using equation 3 for each step $t$.

## 3 SUBSAMPLING ACQUISITION MODEL

The mathematical formulation for the degradation in the acquisition process can be calculated using two models: one for modeling acquisition in images represented as vectors, and the second model for images represented as 2-D arrays.

### 3.1 VECTOR MATHEMATICAL MODEL

In mathematical terms, an image (a dense or complete seismic shot-gather) can be denoted as a vector $\mathbf{x} \in \mathbb{R}^{MN}$. The measurements or subsampled acquisition are referred to as $\mathbf{y} \in \mathbb{R}^{MN}$, where zero values represent the missing traces defined by a vector $\mathbf{d}$. The following equation describes the acquisition process:

$$\mathbf{y} = \mathbf{H_D}\mathbf{x}, \tag{5}$$

where $\mathbf{H_D} \in \mathbb{R}^{MN \times MN}$ simulates the acquisition process, defined as $\mathbf{H_D} = \mathbf{D} \otimes \mathbf{I}$, $\otimes$ represents the Kronecker product, $\mathbf{D} \in \mathbb{R}^{N \times N}$ is a diagonal matrix version of $\mathbf{d}$ such that $\mathbf{D}_{i,j} = 0 \ \forall i, j \in \mathbf{d} \ \wedge \forall i, j \in \{1, 2, \ldots, N\} \ \wedge i \neq j$, where $\mathbf{d}$ indicates the zero values in the diagonal of $\mathbf{H_D}$, and $\mathbf{I} \in \mathbb{R}^{M \times M}$ is an identity matrix. Representing the problem thereby enables the use of linear algebra tools for gradient calculation and iterative updates (Liu et al., 2015; Yang et al., 2012; Kazemi et al., 2016).

### 3.2 MATRIX MATHEMATICAL MODEL

A two-dimensional array $\mathbf{X} \in \mathbb{R}^{M \times N}$ mathematically represents a 2-D dense or complete seismic shot-gather. Similarly, the measurements are referred to as $\mathbf{Y} \in \mathbb{R}^{M \times N}$, where zero columns represent the missing traces. The following equation describes the acquisition process:

$$\mathbf{Y} = \mathbf{H} \odot \mathbf{X}, \tag{6}$$

where $\odot$ symbolizes the element-wise product and $\mathbf{H} = [\mathbf{h}_1, \mathbf{h}_2, \mathbf{h}_j, ..., \mathbf{h}_n] \in \{0, 1\}^{M \times N}$ is the mask that simulates the acquisition process with $\mathbf{h}_j = 0 \ \forall j \in \mathbf{d}$ indicating the $j$th column of $\mathbf{H}$ with zeros. The purpose of this binary mask operator is to model the sampling process in unsupervised DL frameworks, where the acquisition is represented as an element-wise product(Rodríguez-López et al., 2023; Liu et al., 2021).

## 4 PROPOSED METHOD

The proposed method for seismic data reconstruction using a Restoration Operator based on a Deep Image Prior (DM-RODIP), illustrated in Figure 1, adapts the restoration operator by Hu et al. (2023), as it has demonstrated strong performance in various image restoration tasks, to guide the reverse denoising process of the diffusion model. This approach uses a pre-trained deep learning model to condition a proximal operator; specifically, in our case, we employ an Attention U-Net trained following the traditional DIP approach while executing the DM reverse process.

The reconstruction task involves estimating $\mathbf{X}$ from $\mathbf{Y}$. To this end, the first stage of our method consists of training a DM model to generate images $\mathbf{X}$, while the second stage designs a solver to guide the reverse reconstruction process.

The DM receives a noisy sample $\mathbf{X}_t^*$ from the previous step and predicts an unconditional estimate $\mathbf{X}_{t-1}$. The solver then generates an estimated solution $\hat{\mathbf{X}}_0$ to the reconstruction problem, conditioned on the DM's prediction $\mathbf{X}_{t-1}$:

$$\mathbf{X}_{t-1} = \frac{1}{\sqrt{\alpha_t}}\left(\mathbf{X}_t^* - \frac{\beta_t}{\sqrt{1 - \bar{\alpha}_t}}\boldsymbol{\epsilon}_\theta(\mathbf{X}_t^*, t)\right) + \sqrt{\beta_t}\boldsymbol{\epsilon}. \tag{7}$$

At the next reverse step, $t - 1$, the noisy sample $\mathbf{X}_t^*$ is computed from the current estimate $\hat{\mathbf{X}}_0$ via the forward process.

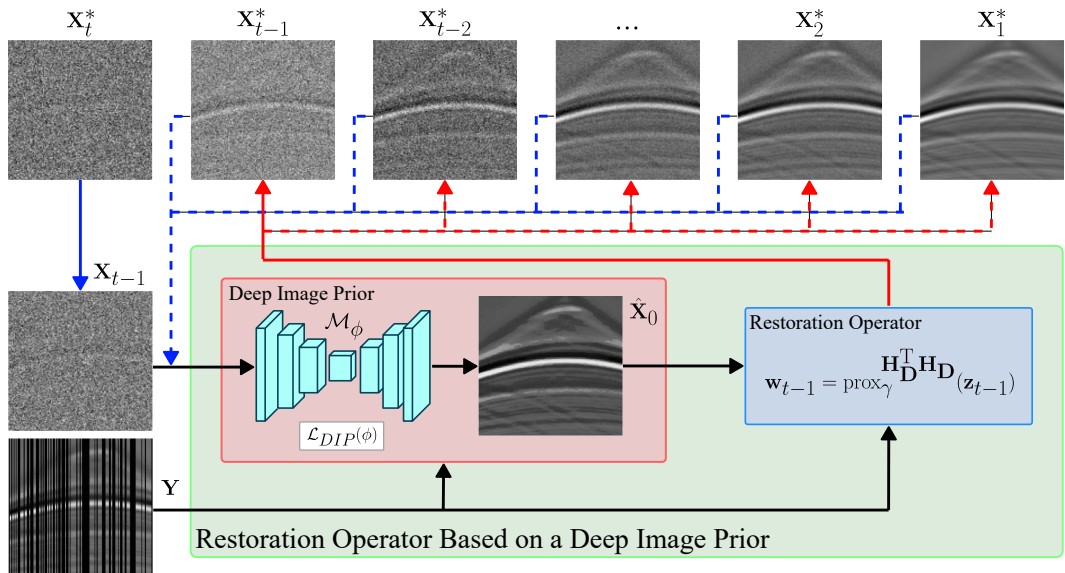

Figure 1: Scheme of the proposed method using a Restoration Operator and a Deep Image Prior to guide the reverse diffusion process for seismic trace reconstruction. The red arrows represent the forward process in equation 2, and the blue arrows represent the reverse process described in equation 7.

### 4.1 DEEP IMAGE PRIOR

In the Restoration Operator based on a Deep Image Prior, an Attention U-Net $\mathcal{M}_\phi$ follows the architecture proposed by Oktay et al. (2018), enabling both solution acquisition and conditioning of the restoration operator, given the noisy unconditional solution $\mathbf{X}_{t-1}$ from equation 7 using the DIP solution,

$$\hat{\mathbf{X}}_0 = \mathcal{M}_\phi(\mathbf{X}_{t-1}). \tag{8}$$

The optimization problem consists of finding the parameters $\phi^*$ to reduce the loss function $\mathcal{L}_{DIP}(\phi)$ given by the data-fidelity term to reduce the distance between the measurements $\mathbf{Y}$ and the prediction of $\mathcal{M}_\phi$,

$$\phi^* = \underset{\phi}{\arg\min} \frac{1}{MN} \|\mathbf{Y} - \mathbf{H} \odot \mathcal{M}_\phi(\mathbf{X}_{t-1})\|_2^2. \tag{9}$$

### 4.2 RESTORATION OPERATOR

To calculate the solution using the restoration operator, we need to work with vector variables, as it is necessary to describe the acquisition process as a linear model. For this, we vectorized the DIP solution, where $\hat{\mathbf{x}}_0 \in \mathbb{R}^{MN}$ is the vectorized version of $\hat{\mathbf{X}}_0 \in \mathbb{R}^{M \times N}$.

The restoration operator, following the proposed method by Hu et al. (2023), is calculated by the scaled proximal operator $\text{prox}_\gamma^{\mathbf{H}_\mathbf{D}^\mathsf{T}\mathbf{H}_\mathbf{D}}(\mathbf{z}_{t-1})$ given by the least-squares data-fidelity term to obtain a measurement consistency and efficient solution (Kamilov et al., 2023), and the weighted Euclidean semi-norm using the conditioned variable $\mathbf{z}_{t-1}$ which is expressed as:

$$\mathbf{z}_{t-1} = \mathbf{w}_t - \gamma\tau\mathbf{g}, \tag{10}$$

where the constants $\gamma$ and $\tau$ are positive values that control the weight of $\mathbf{g} = \mathbf{w}_t - \hat{\mathbf{x}}_0$, that incorporates the DIP solution by computing the difference between $\mathbf{w}_t \in \mathbb{R}^{MN}$ and $\hat{\mathbf{x}}_0$ in the conditioning variable $\mathbf{z}_{t-1}$, with $\gamma$ also controlling the weight of the data-fidelity term in the proximal operator $\text{prox}_\gamma^{\mathbf{H}_\mathbf{D}^\mathsf{T}\mathbf{H}_\mathbf{D}}(\mathbf{z}_{t-1})$ defined as follows:

$$\mathbf{w}_{t-1} = \underset{\mathbf{w}_t}{\arg\min} \frac{1}{2}\|\mathbf{w}_t - \mathbf{z}_{t-1}\|_{\mathbf{H}_\mathbf{D}^\mathsf{T}\mathbf{H}_\mathbf{D}}^2 + \frac{\gamma}{2}\|\mathbf{y} - \mathbf{H}_\mathbf{D}\mathbf{w}_t\|_2^2, \tag{11}$$

where $\|\mathbf{w}_t - \mathbf{z}_{t-1}\|_{\mathbf{H}_\mathbf{D}^\mathbf{T}\mathbf{H}_\mathbf{D}} := (\mathbf{w}_t - \mathbf{z}_{t-1})^\mathbf{T}\mathbf{H}_\mathbf{D}^\mathbf{T}\mathbf{H}_\mathbf{D}(\mathbf{w}_t - \mathbf{z}_{t-1})$ denotes the weighted Euclidean semi-norm, and to calculate the data-fidelity term, we use the acquisition process modeling by the operator $\mathbf{H}_\mathbf{D}$. When the least-squares data-fidelity term is not convex or $\mathbf{H}_\mathbf{D}^\mathbf{T}\mathbf{H}_\mathbf{D}$ is positive semidefinite, there may be multiple solutions, and the scaling proximal operator returns one of the solutions.

Finally, the matrix version $\mathbf{W}_{t-1} \in \mathbb{R}^{M \times N}$ of the proximal operator solution $\mathbf{w}_{t-1} \in \mathbb{R}^{MN}$ is adjusted to the noise level at step $t - 1$

$$\mathbf{X}_{t-1}^* = \sqrt{\bar{\alpha}_{t-1}}\mathbf{W}_{t-1} + \sqrt{1 - \bar{\alpha}_{t-1}}\epsilon, \quad \epsilon \sim \mathcal{N}(0, 1). \tag{12}$$

In this way, the objective is that the network $\mathcal{M}_\phi$ learns to obtain a solution to the reconstruction problem with the information from the DM and the restoration operator solution for each step $t$, the reconstruction method is summarized in Algorithm 1. The experiments conducted to determine the optimal parameter settings for the proposed method are described in Appendix A.2.

---

**Algorithm 1** DM-RODIP

---

**Require: $\mathbf{H}, \mathbf{H}_\mathbf{D}, \mathbf{Y}, \gamma, \tau, T, \epsilon_{\theta^*}$**
**Ensure: $\mathbf{X}_T^* \sim \mathcal{N}(0, 1), \mathbf{w}_T \sim \mathcal{N}(0, 1)$**
1: **for** $t = T, ..., 1$ **do**
2:     $\epsilon_t \sim \mathcal{N}(0, 1)$
3:     $\mathbf{X}_{t-1} = \frac{1}{\sqrt{\alpha_t}}\left(\mathbf{X}_t^* - \frac{\beta_t}{\sqrt{1-\bar{\alpha}_t}}\epsilon_{\theta^*}(\mathbf{X}_t^*, t)\right) + \sqrt{\Sigma}\epsilon_t$            $\triangleright$ Reverse process
4:     $\phi^* = \underset{\phi}{\operatorname{argmin}} \frac{1}{MN}\|\mathbf{Y} - \mathbf{H} \odot \mathcal{M}_\phi(\mathbf{X}_{t-1}^*)\|_2^2$            $\triangleright$ DIP model
5:     $\hat{\mathbf{X}}_0 = \mathcal{M}_{\phi^*}(\mathbf{X}_{t-1})$
6:     **if** $t > 1$ **then**            $\triangleright$ Restoration operator
7:         $\mathbf{g} = \mathbf{w}_t - \hat{\mathbf{x}}_0$
8:         $\mathbf{z}_{t-1} = \mathbf{w}_t - \gamma\tau\mathbf{g}$
9:         $\mathbf{w}_{t-1} = \operatorname{prox}_\gamma^{\mathbf{H}_\mathbf{D}^\mathbf{T}\mathbf{H}_\mathbf{D}}(\mathbf{z}_{t-1})$            $\triangleright$ Proximal operator
10:         $\mathbf{X}_{t-1}^* = \sqrt{\bar{\alpha}_{t-1}}\mathbf{W}_{t-1} + \sqrt{1-\bar{\alpha}_{t-1}}\epsilon_t$            $\triangleright$ Forward process
11:     **end if**
12: **end for**
    **return** $\hat{\mathbf{X}}_0$

---

# 5 EXPERIMENTAL SETUP

## 5.1 DATASETS

This section describes the pre-stack seismic datasets used to train the diffusion model and perform the reconstruction task.

### 5.1.1 DATASETS FOR DIFFUSION MODEL TRAINING

For DM training, various synthetic and real datasets were collected, considering different acquisition geometries, including 2D and 3D, as well as onshore and marine scenarios. Table 1 summarizes the datasets used for DM training.

### 5.1.2 DATASETS FOR SEISMIC DATA RECONSTRUCTION

For testing the proposed methods, a synthetic and a real dataset were used, which are summarized in Table 2.

## 5.2 MODEL SETUP

### 5.2.1 DIFFUSION MODEL SETUP

The DM used to generate seismic images was initialized from an available, trained model for post-stack seismic data (Goyes-Peñafiel et al., 2025), to leverage the knowledge gained about this type

Table 1: Datasets used for diffusion model training.

| Name | Type | Time Samples | Traces | Shots |
|------|------|--------------|--------|-------|
| Alaska 41-81 | Field | 400 | 96 | 122 |
| Avo Mobil Viking | Field | 800 | 120 | 1001 |
| BPO200 | Field | 2001 | 1201 | 200 |
| C3 1-1200 | Field | 625 | 544 | 68 |
| C3 1201-2395 | Field | 625 | 544 | 43 |
| RL3042VMM | Field | 2501 | 198 | 117 |
| Split Spread Land | Synthetic | 2842 | 128 | 81 |
| Inline Offset Marine | Synthetic | 2842 | 128 | 81 |
| Seam Phase II | Synthetic | 1034 | 100 | 1080 |
| Gas Lens Devito | Synthetic | 554 | 128 | 490 |

Table 2: Datasets used for seismic data reconstruction.

| Name | Type | Time Samples | Traces | Shots |
|------|------|--------------|--------|-------|
| Land Acquisition | Synthetic | 800 | 100 | 1 |
| Stratton 3D Survey (SEG Wiki, 2018) | Field | 1001 | 80 | 18 |

of seismic image and avoid retraining from scratch. It was fine-tuned and trained with the datasets described in Table 2 for 700 epochs using a cosine variance schedule, $\beta_0 = 0$ and $\beta_T = 0.02$, the AdamW optimizer (Loshchilov & Hutter, 2017) with a learning rate of 0.0003, a batch size of 32 and 1000 diffusion steps ($T = 1000$). The detailed evaluation of this configuration is provided in Appendix A.1.

### 5.2.2 DM-RODIP SETUP

The DIP model follows the Attention U-Net scheme proposed by Oktay et al. (2018) and is trained using the Adam optimizer (Kingma & Ba, 2014), a learning rate of $0.0001$ and 2 epochs per reverse step $t$. The proximal operator model was optimized using a conjugate gradient method (Dai & Yuan, 1999), which is employed for optimization due to its efficiency in solving large-scale, sparse, and symmetric positive-definite systems without requiring matrix inversion.

## 6 EXPERIMENTAL RESULTS

We conducted experiments using the datasets described in Table 2 and resized them for our experiments, so that each shot contains 128 time samples and 128 traces. A randomly subsampled acquisition was simulated with a subsampling rate of $50\%$, and compared our proposed method DM-RODIP against the DIP approach introduced by Liu et al. (2021), which was trained for 7000 epochs using the Attention U-Net architecture from Oktay et al. (2018), as well as the CCSeis-DDPM method proposed by Wang et al. (2024b), a diffusion-based reconstruction framework that employs the same diffusion model to generate pre-stack seismic images utilized in our proposed approach.

The peak signal-to-noise ratio (PSNR) and the structural similarity index measure (SSIM) were used to evaluate the reconstruction quality. After 10 realizations, the DM-RODIP method achieved a mean PSNR of 42.3 dB with a standard deviation of 0.7 dB, and a mean SSIM of 0.95 with a standard deviation of 0.01, indicating that all reconstructions generally converge to nearly the same solution. For illustration purposes, Figure 2 presents the reconstruction corresponding to the best PSNR among the 10 realizations, compared with state-of-the-art approaches. Moreover, in this seismic scenario, the proposed method not only outperforms state-of-the-art methods but also demonstrates improved consistency in regions with large amplitude variations, as illustrated in Figure 2.e).

Furthermore, the proposed method achieves superior reconstruction performance compared to state-of-the-art approaches in the field data reconstruction scenario. After 10 realizations, the DM-RODIP

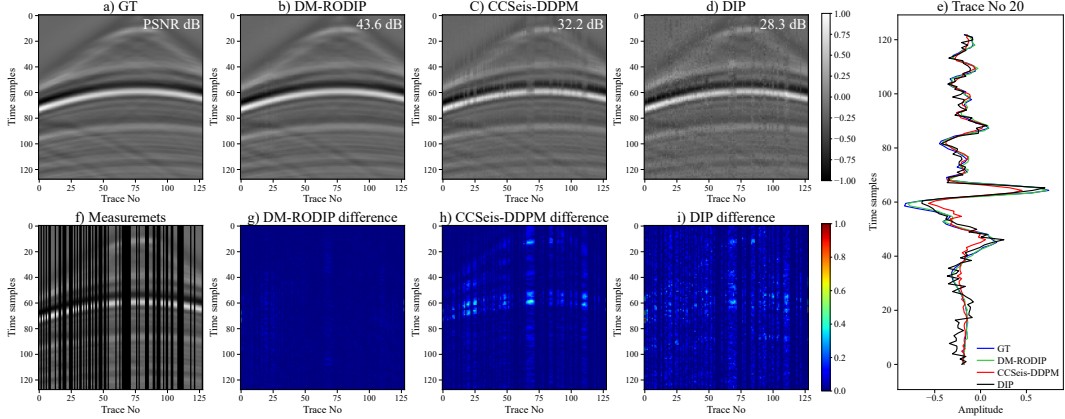

Figure 2: a) Complete Land Acquisition seismic shot-gather or ground truth (GT). Reconstruction with b) the DM-RODIP, c) the CCSeis-DDPM, and d) the DIP methods. e) Comparative between the GT and the reconstructions in trace number 20. f) Measurements, and g)-i) the differences between GT and the respective reconstructions.

method attained a mean PSNR of 31.1 dB with a standard deviation of 0.8 dB and a mean SSIM of 0.79 with a standard deviation of 0.04. For clarity, Figure 3 reports the reconstruction corresponding to the best PSNR among the 10 realizations on Stratton 3D data. In this case, the proposed method provides more accurate reconstructions, particularly in deeper regions and seismic events with limited amplitude variations, where other methods tend to lose structural consistency.

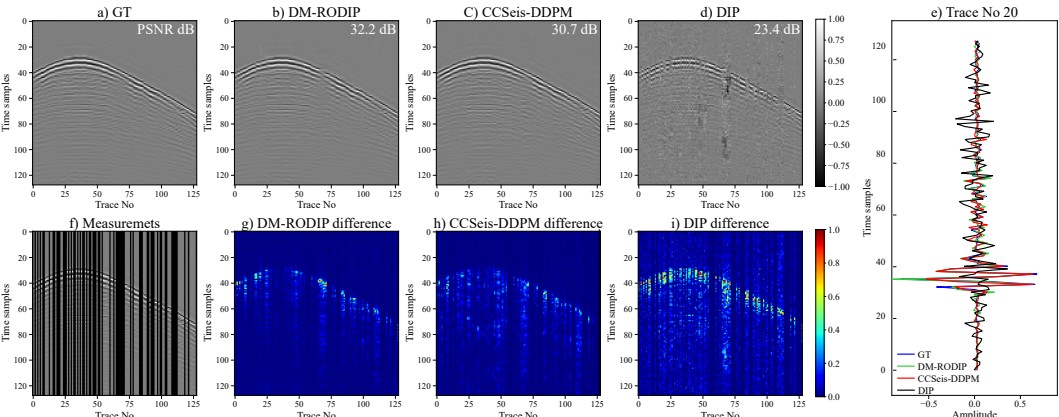

Figure 3: a) Complete Stratton 3D seismic shot-gather or ground truth (GT). Reconstruction with b) the DM-RODIP, c) the CCSeis-DDPM, and d) the DIP methods. e) Comparative between the GT and the reconstructions in trace number 52. f) Measurements, and g)-i) the differences between GT and the corresponding reconstructions.

Table 3 clearly highlights the superiority of the proposed DM-RODIP method over state-of-the-art approaches in terms of PSNR and SSIM. For synthetic data, DM-RODIP surpasses the DIP and CCSeis-DDPM methods by 10.2 dB in PSNR and 0.09 in SSIM, while for field data it achieves gains of 1.0 dB in PSNR and 0.04 in SSIM. These quantitative improvements confirm the consistent advantage of the proposed method, demonstrating not only its superiority in reconstruction accuracy but also its robustness across synthetic and real seismic acquisition scenarios.

Additionally, to further evaluate performance across different acquisition conditions, Table 4 summarizes the results of the DM-RODIP method under various subsampling rates in synthetic Land Acquisition data. The results show that the proposed method exhibits strong robustness to changes in subsampling density, consistently preserving high reconstruction quality even when a significant

Table 3: Performance assessment on Land Acquisition and Stratton 3D datasets for the DM-RODIP, CCSeis-DDPM, and DIP methods using PSNR and SSIM metrics.

| Dataset | Method | PSNR (dB) ↑ | SSIM ↑ |
|---|---|---|---|
| | DM-RODIP | **42.3 ± 0.7** | **0.95 ± 0.01** |
| Land Acquisition | CCSeis-DDPM | 32.1 ± 0.07 | 0.86 ± 0.01 |
| | DIP | 24.6 ± 2.3 | 0.53 ± 0.11 |
| | DM-RODIP | **31.1 ± 0.8** | **0.79 ± 0.04** |
| Stratton 3D | CCSeis-DDPM | 30.1 ± 0.35 | 0.75 ± 0.01 |
| | DIP | 25.1 ± 1.7 | 0.14 ± 0.07 |

number of traces are omitted during acquisition. This robustness highlights the ability of DM-RODIP to deliver stable and accurate reconstructions under challenging acquisition scenarios, reinforcing its potential for practical seismic applications. Further experiments with different subsampling patterns and rates are presented in Appendix A.3, and reconstruction tests on data contaminated with various levels of noise are provided in Appendix A.4.

Table 4: PSNR and SSIM metrics for different subsampling rates in the synthetic Land Acquisition data reconstruction scenario.

| Subsampling Rate | PSNR (dB) ↑ | SSIM ↑ |
|---|---|---|
| 20% | 42.4 | 0.97 |
| 30% | 42.4 | 0,97 |
| 40% | 42.3 | 0.96 |
| 50% | 42.3 | 0.95 |
| 60% | 39.1 | 0.90 |

## 7 CONCLUSIONS AND FUTURE WORK

The DM-RODIP method was proposed for seismic trace reconstruction, utilizing only the measurements. The proposed method reduces computational costs by eliminating the need to re-train the diffusion model for each experiment with dense and uniform acquisitions. In contrast to state-of-the-art DM-based methods, it achieves superior reconstruction in synthetic and field data while maintaining robustness across different subsampling rates. As a potential application, this approach could be extended to other imaging restoration problems, where our mask can be replaced by another degradation operator depending on the task, inpainting, super-resolution, blurring, etc.

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

# A APPENDIX

## A.1 DIFFUSION MODEL RESULTS

The evaluation was performed by initializing the parameters $\theta$ of the DM $\epsilon_\theta$ either randomly or with the weights of the DM trained on post-stack seismic images by Goyes-Peñafiel et al. (2025). For the FID metric, we used the dataset described in Table 1 as the source of real (reference) images to assess whether training enabled the model to capture the distribution of pre-stack seismic images.



Figure 4: Seismic images generated by the DM initialized with weights from a DM trained on post-stack seismic images.

Figure 4 presents the results obtained by initializing the DM with weights from a model trained on post-stack seismic images, while Table 5 reports the outcomes obtained by generating 500 images with each trained DM. The results demonstrate that initializing the DM with pre-trained weights from post-stack seismic data leads to superior performance in terms of both FID and IS, confirming the improved capability of the model to generate high-quality and diverse pre-stack seismic images.

Table 5: Comparison of FID and IS scores for different DM.

| Parameter Initialization | FID | IS |
|---|---|---|
| Randomly | 1.23 | 2.98 |
| Diffusion Model for Post-stack Images | **0.29** | **2.38** |

## A.2 RESTORATION OPERATOR BASED ON A DEEP IMAGE PRIOR PARAMETER SETTING

Table 6 presents the evaluation of the DM-RODIP algorithm, where we varied the initialization of the variable $\mathbf{w}_T$ as well as the values of the constants $\gamma$ and $\tau$ in the proximal operator. The PSNR and SSIM values reported correspond to the mean over 10 repeated experiments using the Land Acquisition dataset described in Table 2, with a randomly subsampled acquisition simulated at a rate of 0.5. It can be observed that for each parameter setting, the reconstruction quality was quantified using the proposed metrics, with the best-performing configuration highlighted in bold. Based on this evaluation, the optimal configuration corresponds to 1000 DM steps, 2 epochs in the DIP model, initialization of $\mathbf{w}_T$ as Gaussian random noise, and parameter values of $\gamma = 0.1$ and $\tau = 0.1$.

## A.3 RECONSTRUCTION RESULTS UNDER DIFFERENT SUBSAMPLING TYPES AND RATES

Table 7 presents the results across various scenarios with Land Acquisition data and Figure 5 illustrate the reconstruction results at a subsampling rate of 0.5 using different sampling types. The

Figure 5: Comparative results with random, jitter, and uniform subsampling between complete synthetic seismic shot-gather or ground truth (GT), reconstruction with the DM-RODIP, the DM-DIGP, the CCSeis-DDPM, and the DIP methods. Measurements and the differences between GT and the respective reconstructions.

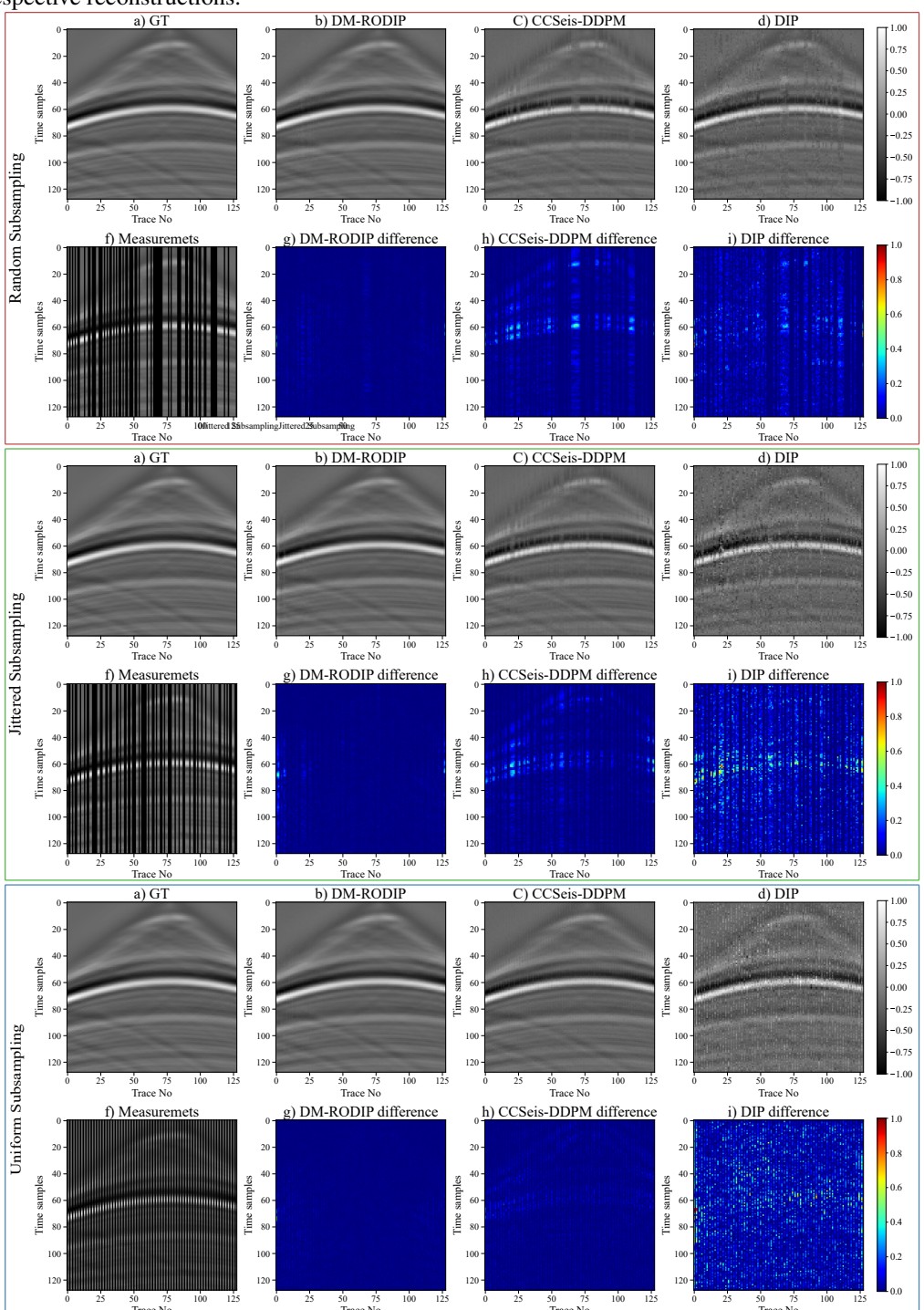

reported PSNR and SSIM values correspond to the mean over 10 independent realizations. After 10 realizations, the DM-RODIP method achieves superior results in different scenarios in terms of SSIM and PSNR.

Table 6: Evaluation of different initializations and proximal operator parameters.

| PSNR (dB) | SSIM | Steps | Epochs | Initialization $\mathbf{w}_T$ | $\gamma$ | $\tau$ |
|---|---|---|---|---|---|---|
| 32.6 | 0.82 | 1000 | 2 | Ones | 0.1 | 0.1 |
| 32.6 | 0.81 | 1000 | 2 | Zeros | 0.1 | 0.1 |
| 38.4 | 0.98 | 1000 | 2 | Gaussian Noise | 0.1 | 0.5 |
| 37.2 | 0.87 | 1000 | 2 | Gaussian Noise | 0.1 | 0.05 |
| 39.1 | 0.91 | 1000 | 2 | Gaussian Noise | 0.5 | 0.1 |
| 40.2 | 0.94 | 1000 | 2 | Gaussian Noise | 0.05 | 0.1 |
| **42.3** | **0.96** | 1000 | 2 | Gaussian Noise | 0.1 | 0.1 |
| 34.1 | 0.85 | 500 | 2 | Gaussian Noise | 0.1 | 0.1 |

Table 7: Comparison of PSNR and SSIM values for different reconstruction methods at various subsampling rates and types.

| Subsampling Rate | Subsampling Type | DM-RODIP | | CCSeis-DDPM | | DIP | |
|---|---|---|---|---|---|---|---|
| | | PSNR (dB) | SSIM | PSNR (dB) | SSIM | PSNR (dB) | SSIM |
| 20% | Random | **42.4** | **0.97** | 35.6 | 0.93 | 32.5 | 0.78 |
| | Jittered | **43.8** | **0.96** | 38.4 | 0.95 | 33.3 | 0.80 |
| 30% | Random | **42.4** | **0.97** | 34.5 | 0.91 | 29.3 | 0.75 |
| | Jittered | **43.6** | **0.96** | 37.1 | 0.94 | 30.1 | 0.71 |
| 40% | Random | **42.3** | **0.96** | 33.3 | 0.88 | 26.2 | 0.69 |
| | Jittered | **42.8** | **0.94** | 35.9 | 0.92 | 27.2 | 0.64 |
| 50% | Uniform | **41.9** | **0.93** | 37.4 | 0.91 | 23.8 | 0.47 |
| | Random | **42.3** | **0.95** | 32.1 | 0.86 | 24.6 | 0.53 |
| | Jittered | **38.3** | **0.86** | 32.8 | 0.88 | 24.2 | 0.52 |

## A.4 RECONSTRUCTION RESULTS UNDER DIFFERENT NOISE LEVELS

Noise is an inherent characteristic of acquired seismic data. Several common noise processes are well approximated by Gaussian statistics and manifest across multiple scales. This complicates reconstruction, as the model must recover the underlying wavefield while simultaneously separating it from the corrupting component. To evaluate this challenge under controlled conditions, we construct a synthetic scenario in which a seismic signal is perturbed by Gaussian noise at level $\rho$, yielding $\mathbf{Y}_\rho$, and subsequently subsampled at $30\%$ using jittered subsampling to produce $\mathbf{Y}$ according to the subsampling acquisition model.

Table 8 reports performance across noise levels in terms of PSNR and SNR, enabling a quantitative comparison of reconstruction quality; reconstruction PSNR relative to the clean target remains high and degrades gracefully as noise increases, while PSNR and SNR measured against the noisy inputs confirm substantial denoising and data fidelity preservation across all $\rho$ levels.

Table 8: Reconstruction performance across different noise levels $\rho$ in terms of PSNR and SNR (dB) against the clean target $\mathbf{X}$ and the noisy signal $\mathbf{Y}_\rho$.

| $\rho$ | SNR($\mathbf{Y}_\rho$, $\mathbf{X}$) (dB) | PSNR($\hat{\mathbf{X}}$, $\mathbf{X}$) (dB) | PSNR($\hat{\mathbf{X}}$, $\mathbf{Y}_\rho$) (dB) | SNR($\hat{\mathbf{X}}$, $\mathbf{X}$) (dB) |
|---|---|---|---|---|
| 0.8 | 7.97 | 25.7 | 49.5 | 3.56 |
| 0.7 | 8.99 | 26.6 | 48.9 | 5.43 |
| 0.6 | 10.19 | 27.7 | 48.0 | 7.81 |
| 0.5 | 11.65 | 29.1 | 46.9 | 7.29 |
| 0.4 | 13.47 | 30.3 | 43.2 | 5.56 |

As $\rho$ decreases (higher input SNR), PSNR($\hat{\mathbf{X}}$, $\mathbf{X}$) improves from 25.1 to 30.3 dB around mid-range noise and remains stable at lower noise, while SNR($\hat{\mathbf{X}}$, $\mathbf{X}$) (dB) stays within 3.56–7.81 dB, indicating effective denoising and fidelity preservation across noise regimes; additionally, the consistently

high PSNR($\hat{\mathbf{X}}$, $\mathbf{Y}_\rho$) (dB) underscores robustness with respect to the noisy reference, supporting the method's reliability under varying noise levels as showing in Figure 6.

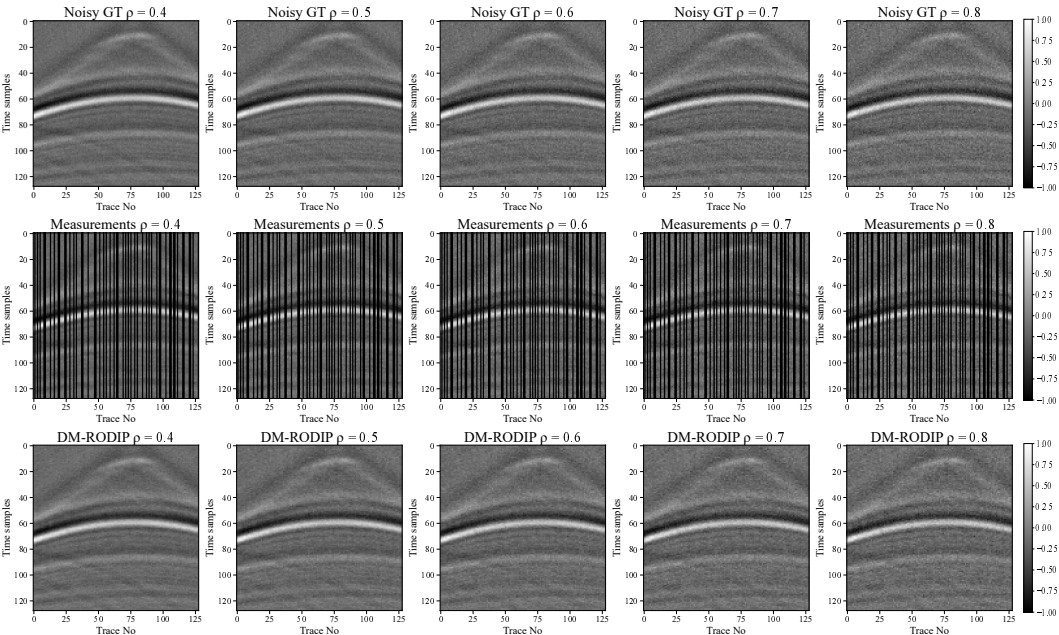

Figure 6: DM-RODIP reconstruction with different noise levels $\rho$.

### A.5 THE USE OF LARGE LANGUAGE MODELS

Large Language Models (LLMs), such as ChatGPT/Claude, were used solely to improve writing quality and to draft table layouts. With LLM assistance, the authors received grammar, style, and clarity suggestions on author-written text, and all edits were reviewed and either accepted or rejected by the authors. LLMs also assisted in drafting table structures and captions from author-provided results; every table was cross-checked against the original data and finalized by the authors. No LLM-generated text, data, analyses, code, or references were accepted without human verification, and the authors take full responsibility for the paper's content.

