# OpenReview forum: "Constrained Probabilistic Diffusion Model for Seismic Data Reconstruction Using a Restoration Operator Based on a Deep Image Prior"
_ICLR.cc/2026/Conference — Submitted to ICLR 2026_

### Official Review · Reviewer_Azzx · 2025-10-23

**Soundness:** 2
**Presentation:** 3
**Contribution:** 2
**Rating:** 2
**Confidence:** 3

**Summary:**

This paper introduces DM-RODIP, a method for seismic data reconstruction that integrates a diffusion model (DM) with a Restoration Operator based on a Deep Image Prior (DIP). The key idea is to guide the reverse diffusion process using a pretrained restoration network (an Attention U-Net trained in DIP fashion), thereby avoiding full retraining of the diffusion model for each reconstruction task. The method aims to reduce computational cost of retraining while improving reconstruction quality on both synthetic and field seismic datasets. Experiments demonstrate performance  gains over  baselines such as CCSeis-DDPM and standalone DIP reconstructions.

**Strengths:**

* Reduced computational overhead through reuse of pretrained diffusion models
The main practical contribution lies in reusing a pretrained diffusion model and optimizing only a lightweight Deep Image Prior (DIP) restoration network for each reconstruction task. This design potentially lowers GPU cost, training time, and data requirements compared to full retraining of diffusion models.

* The proposed method seems to outperform the baselines and improve the quality of reconstruction.

**Weaknesses:**

* The conceptual novelty is limited. The proposed framework primarily combines existing components — a pretrained diffusion model, a Deep Image Prior, and a restoration prior— without introducing a new theoretical idea or methodological principle. There is no additional insight as to why this combination achieves better performance.

*  Contribution is ambiguous. The paper frames the method as a new diffusion variant (“Constrained Probabilistic Diffusion Model”), but in practice it is just  reusing a pretrained DM with a DIP-based refinement. The key contribution therefore lies in connecting previously established ideas rather than advancing diffusion modeling itself.

* The paper  lack of theoretical or empirical analysis of the DRP.  The paper does not analyze the goal and role of the  restoration operator in  the diffusion process. The guidance from the restoration operator is not clear.

* The scale of the experiments is limited. This conceptually should work on other inverse problems, however, no other image restoration or inverse problem is discussed.

**Questions:**

How does including the DRP affect the computation cost and performance?
How does the restoration operator mathematically constrain or modify the reverse diffusion dynamics? Is there an objective function or probabilistic interpretation linking the DIP output to the diffusion posterior?

---

> ### Author Response · Authors · 2025-11-27
>
> Thank you for your feedback. To address the aforementioned weaknesses, we are preparing a new section titled "Probabilistic Interpretation of DM-RODIP", in which the probabilistic formulation of the proposed method will be detailed; this section will include the following:
>
> The reconstruction variable is a seismic shot-gather $\mathbf{X} \in \mathbb{R}^{M \times N}$, and the measurements $\mathbf{Y} \in \mathbb{R}^{M \times N}$ are obtained through the acquisition model in Eqs. (5)–(6) with additive Gaussian noise.
> For the matrix model in Eq. (6), we assume
> $$
> \mathbf{Y} = \mathbf{H} \odot (\mathbf{X} + \eta), \qquad \eta \sim \mathcal{N}\bigl(0,\sigma^2 \mathbf{I}\bigr),
> $$
>
> so that the likelihood associated with the measurement model is
>
> $$
> p(\mathbf{Y} \mid \mathbf{X}) \propto \exp \left(  - \frac{1}{2\sigma^2}\, \| \mathbf{Y} - \mathbf{H} \odot \mathbf{X} \|_F^2 \right),
> $$
>
> which coincides with the quadratic data-fidelity term used in Eqs. (9) and (11) (up to a scaling factor).
>
> In DM-RODIP, the posterior distribution implicitly targeted at reverse step $t$ can be written as
>
> $$\tilde{p}t(\mathbf{X} \mid \mathbf{Y}) \propto p{\text{DM}}(\mathbf{X})p_{\text{DIP},t}(\mathbf{X} \mid \mathbf{Y})p_{\text{lik},t}(\mathbf{X} \mid\mathbf{Y})$$
>
> where:
> (i) $p_{\text{DM}}(\mathbf{X})$ is the prior learned by the diffusion model over seismic images,
> (ii) $p_{\text{DIP},t}(\mathbf{X} \mid \mathbf{Y})$ is an implicit, data-conditioned prior induced by the DIP at step $t$, and
> (iii) $p_{\text{lik},t}(\mathbf{X} \mid \mathbf{Y})$ is a Moreau-smoothed likelihood term derived from the proximal operator associated with the quadratic data fidelity in Eqs. (5)–(6).
> Equivalently, the solver performs stochastic ascent on the time-varying log-density
>
> $\log \tilde{p}\_{t} (\mathbf{X} \mid \mathbf{Y}) = \log p\_{\text{DM}}(\mathbf{X})+ \log p\_{\text{DIP},t} (\mathbf{X} \mid \mathbf{Y})+ \log p\_{\text{lik},t} (\mathbf{X} \mid \mathbf{Y}) + C\_t,$
>
> with $C_t$ absorbing all constants independent of $\mathbf{X}$.
>
> In the DM-RODIP solver, each reverse step combines:
> (i) a denoising update driven by the diffusion score (Eq. (7)),
> (ii) a DIP pull toward a data-conditioned reference (Eq. (10)), and
> (iii) a proximal correction enforcing data consistency through a Moreau-smoothed likelihood (Eq. (11)),
> followed by re-noising according to Eq. (12).
>
> This sequence is equivalent to a stochastic ascent step on $\log \tilde{p}_t(\mathbf{X} \mid \mathbf{Y})$, where the diffusion score encodes the seismic data prior, the proximal operator encodes the measurement model through a Moreau-smoothed quadratic fidelity, and the DIP acts as an implicit, data-conditioned prior around the DIP prediction.
>
> Consequently, the iterations drift toward regions that are simultaneously high probability under the learned seismic distribution, close to the DIP manifold favored by the measurements, and consistent with the acquisition geometry, all without retraining the diffusion model for each operator.

---

> > ### Author Response · Authors · 2025-11-27
> >
> > Regarding your questions, we would like to comment on the following:
> >
> > ### How does including the DRP affect the computation cost and performance?
> >
> > Thank you for that appreciation. To determine the impact of the DRP, an evaluation was performed using only the diffusion model and the proximal operator (DM + DRP) and the diffusion model plus DIP (DM + DIP). The reconstruction task was evaluated using synthetic data Land Acquisition in a random subsampling scenario with a 50\% sampling rate. In terms of PSNR and SSIM, the DM + Restoration experiment achieved a mean PSNR of 15.8 dB and an SSIM of 0.11, while the DM + DIP experiment achieved a mean PSNR of 40.2 dB and an SSIM of 0.94. The final method (DM + DIP + DRP) obtains the best results, with a mean PSNR of 42.3 dB and an SSIM of 0.95, thus demonstrating the importance of the combination of the DRP and DIP modules. The computational time using only DM + DIP is 192 seconds, while the proposed method (DM + DIP + DRP) requires 1543 seconds. However, since adding DRP provides a gain of 2.1 dB in PSNR, this increase in computational cost is considered acceptable for this scenario.}
> >
> > ### How does the restoration operator mathematically constrain or modify the reverse diffusion dynamics?
> >
> > The proximal update in Eq.(11) realizes the force of the data consistency as the gradient of a Moreau‑smoothed quadratic data fidelity (for the acquisition model in Eqs.(5)–(6)), yielding a stable surrogate of the likelihood gradient that reduces to a closed‑form projection at measurements. Concretely, at each reverse step we first obtain a denoised estimate (via Eq.(7)), then apply a DIP pull (Eq.(10)) and a proximal correction (Eq.(11)) before re‑noising (Eq.(12)). This sequence is equivalent to taking a stochastic ascent step on a time‑varying probabilistic energy whose terms are: the diffusion score (data distribution), a Moreau‑smoothed data fidelity (measurement model), and a quadratic attraction around the DIP prediction (implicit, data‑conditioned prior).
> >
> > ### Is there an objective function or probabilistic interpretation linking the DIP output to the diffusion posterior?
> >
> > The Deep Image Prior (DIP) provides an instance‑conditioned reference $\hat{\mathbf{X}}\_{0} = \mathcal{M}\_\theta(\mathbf{X}\_{t-1})$ obtained by minimizing the data term in Eq. (9), which acts as a solution guidance toward reconstructions that are both plausible and consistent with the measurements.
> >
> >
> > Finally, if the document is accepted, the corresponding adjustments will be made to address the weaknesses and questions raised, including adjustments to the graphs to add the new results.

---

### Official Review · Reviewer_xbj5 · 2025-10-28

**Soundness:** 3
**Presentation:** 3
**Contribution:** 3
**Rating:** 6
**Confidence:** 4

**Summary:**

This paper proposes DM-RODIP, a constrained probabilistic diffusion framework for seismic data reconstruction under sparse and irregular sampling conditions. The method integrates three components: a pre-trained diffusion model (DM) serving as a probabilistic prior of seismic data distribution, a Deep Image Prior (DIP) module acting as an implicit structural regularizer, and a Restoration Operator formulated as a proximal optimization step enforcing measurement consistency. Unlike conventional diffusion-based reconstruction methods that require retraining the entire model for each dataset or sampling configuration, DM-RODIP achieves efficient reconstruction without retraining, by coupling DIP and Restoration into the reverse diffusion loop. Experiments on synthetic and field datasets (Land 3D, Stratton) show consistent improvements over DIP-only and DDPM-based baselines.

**Strengths:**

1. The paper elegantly combines diffusion models (probabilistic priors) with proximal optimization and implicit deep priors. The Restoration Operator enforces physical measurement constraints, bridging the gap between generative modeling and data-consistent inversion. The overall pipeline is theoretically sound and computationally efficient.

2. Unlike previous DM-based methods that are highly data-specific, this approach decouples distribution learning (offline) from reconstruction (online), allowing direct adaptation to new observation patterns.

3. Visual examples convincingly show better layer continuity and fewer artifacts in reconstructed seismic sections. The method is applicable not only to seismic reconstruction but also to other ill-posed imaging problems (denoising, super-resolution, inpainting).

**Weaknesses:**

1. The paper lacks quantitative evidence showing the distinct contributions of DIP and the Restoration Operator. For instance, results with “DM only”, “DM + Restoration”, and full “DM + DIP + Restoration” would help clarify each module’s necessity.
2. The paper describes the DIP–Restoration coupling intuitively but lacks a formal Bayesian or probabilistic justification.
3. The workflow diagram (Fig. 1) is not intuitive. It illustrates the iterative reverse diffusion loop but does not clearly differentiate between the training phase (DM pre-training) and the inference/reconstruction phase. I recommend redesigning the figure to explicitly show two parts: (1) Offline training of the diffusion model using full seismic data; (2) Online reconstruction using DIP and Restoration Operator on observed sparse data.
4. The method is compared mainly with DIP and DDPM; missing stronger baselines such as conditional or plug-and-play diffusion models.
5. Several notations are undefined or underexplained.​

**Questions:**

1. Can you provide quantitative evidence (PSNR/SSIM) showing the effect of removing the DIP module?
2. How sensitive is reconstruction performance to hyper parameters in the Restoration Operator?
3. Is the Restoration Operator applied at every diffusion step or intermittently?

---

> ### Author Response · Authors · 2025-11-27
>
> Thank you for your feedback. To address the aforementioned weaknesses, we are preparing a new section titled "Probabilistic Interpretation of DM-RODIP", in which the probabilistic formulation of the proposed method will be detailed; this section will include the following:
>
> The reconstruction variable is a seismic shot-gather $\mathbf{X} \in \mathbb{R}^{M \times N}$, and the measurements $\mathbf{Y} \in \mathbb{R}^{M \times N}$ are obtained through the acquisition model in Eqs. (5)–(6) with additive Gaussian noise.
> For the matrix model in Eq. (6), we assume
> $$
> \mathbf{Y} = \mathbf{H} \odot (\mathbf{X} + \eta), \qquad \eta \sim \mathcal{N}\bigl(0,\sigma^2 \mathbf{I}\bigr),
> $$
>
> so that the likelihood associated with the measurement model is
>
> $$
> p(\mathbf{Y} \mid \mathbf{X}) \propto \exp \left(  - \frac{1}{2\sigma^2}\, \| \mathbf{Y} - \mathbf{H} \odot \mathbf{X} \|_F^2 \right),
> $$
>
> which coincides with the quadratic data-fidelity term used in Eqs. (9) and (11) (up to a scaling factor).
>
> In DM-RODIP, the posterior distribution implicitly targeted at reverse step $t$ can be written as
>
> $$\tilde{p}t(\mathbf{X} \mid \mathbf{Y}) \propto p{\text{DM}}(\mathbf{X})p_{\text{DIP},t}(\mathbf{X} \mid \mathbf{Y})p_{\text{lik},t}(\mathbf{X} \mid\mathbf{Y})$$
>
> where:
> (i) $p_{\text{DM}}(\mathbf{X})$ is the prior learned by the diffusion model over seismic images,
> (ii) $p_{\text{DIP},t}(\mathbf{X} \mid \mathbf{Y})$ is an implicit, data-conditioned prior induced by the DIP at step $t$, and
> (iii) $p_{\text{lik},t}(\mathbf{X} \mid \mathbf{Y})$ is a Moreau-smoothed likelihood term derived from the proximal operator associated with the quadratic data fidelity in Eqs. (5)–(6).
> Equivalently, the solver performs stochastic ascent on the time-varying log-density
>
> $\log \tilde{p}\_{t} (\mathbf{X} \mid \mathbf{Y}) = \log p\_{\text{DM}}(\mathbf{X})+ \log p\_{\text{DIP},t} (\mathbf{X} \mid \mathbf{Y})+ \log p\_{\text{lik},t} (\mathbf{X} \mid \mathbf{Y}) + C\_t,$
>
> with $C_t$ absorbing all constants independent of $\mathbf{X}$.
>
> In the DM-RODIP solver, each reverse step combines:
> (i) a denoising update driven by the diffusion score (Eq. (7)),
> (ii) a DIP pull toward a data-conditioned reference (Eq. (10)), and
> (iii) a proximal correction enforcing data consistency through a Moreau-smoothed likelihood (Eq. (11)),
> followed by re-noising according to Eq. (12).
>
> This sequence is equivalent to a stochastic ascent step on $\log \tilde{p}_t(\mathbf{X} \mid \mathbf{Y})$, where the diffusion score encodes the seismic data prior, the proximal operator encodes the measurement model through a Moreau-smoothed quadratic fidelity, and the DIP acts as an implicit, data-conditioned prior around the DIP prediction.
>
> Consequently, the iterations drift toward regions that are simultaneously high probability under the learned seismic distribution, close to the DIP manifold favored by the measurements, and consistent with the acquisition geometry, all without retraining the diffusion model for each operator.
>
> Additionally, the DPS method was evaluated, and the results of this method were added to Table 3.
>
> **Table 3.** Performance assessment on Land Acquisition and Stratton 3D datasets for the DM-RODIP, CCSeis-DDPM, DPS, and DIP methods using PSNR and SSIM metrics.
>
> | Method       | Dataset         | PSNR (dB) ↑      | SSIM ↑          | Time (s) ↓        |
> |-------------|-----------------|------------------|-----------------|-------------------|
> | DM-RODIP    | Land Acquisition| **42.3 ± 0.7**   | **0.95 ± 0.01** | 1543 (35 + 1508)  |
> |             | Stratton 3D     | **31.1 ± 0.8**   | **0.79 ± 0.04** |                   |
> | CCSeis-DDPM | Land Acquisition| 32.1 ± 0.07      | 0.86 ± 0.01     | 364 (35 + 329)    |
> |             | Stratton 3D     | 30.1 ± 0.35      | 0.75 ± 0.01     |                   |
> | DPS         | Land Acquisition| 20.2 ± 0.02      | 0.29 ± 0.01     | 69 (35 + 34)      |
> |             | Stratton 3D     | 29.3 ± 0.01      | 0.78 ± 0.01     |                   |
> | DIP         | Land Acquisition| 24.6 ± 2.3       | 0.53 ± 0.11     | **125**           |
> |             | Stratton 3D     | 25.1 ± 1.7       | 0.14 ± 0.07     |                   |
>
> As shown, the proposed method outperforms the DPS method. This is expected as DPS is designed for restoration tasks in natural images. In our scenario, the problem corresponds to an inpainting setting in which entire columns of the image are lost. Consequently, methods that are not focused on this type of degradation exhibit limited performance in reconstructing seismic traces.

---

> > ### Author Response · Authors · 2025-11-27
> >
> > Regarding your questions, we would like to comment on the following:
> >
> > ### Can you provide quantitative evidence (PSNR/SSIM) showing the effect of removing the DIP module?
> >
> > Thank you for the recommendation. To determine the impact of the DIP module, an evaluation was performed using only the diffusion model and the proximal operator (DM + Restoration) and the diffusion model plus DIP (DM + DIP). The reconstruction task was evaluated using synthetic data in a random subsampling scenario with a 50% sampling rate. In terms of PSNR and SSIM, the DM + Restoration experiment achieved a PSNR of 15.8 dB and an SSIM of 0.11, while the DM + DIP experiment achieved a PSNR of 40.2 dB and an SSIM of 0.94, thus demonstrating the crucial role of the DIP module in conditioning the proximal operator.
> >
> > ### How sensitive is reconstruction performance to hyper parameters in the Restoration Operator?
> >
> > Table 6 shows the model's sensitivity to the adjustment of these parameters, which are applied to both synthetic and real data. As shown, changes to these constants have a substantial impact on reconstruction quality.
> >
> > **Table 6.** Evaluation of different initializations and proximal operator parameters.
> >
> > | PSNR (dB) ↑ | SSIM ↑ | Steps | Epochs | Initialization $\mathbf{w}_T$ | $\gamma$ | $\tau$ |
> > |-------------|--------|-------|--------|-------------------------------|----------|--------|
> > | 32.6        | 0.82   | 1000  | 2      | Ones                          | 0.1      | 0.1    |
> > | 32.6        | 0.81   | 1000  | 2      | Zeros                         | 0.1      | 0.1    |
> > | 38.4        | 0.98   | 1000  | 2      | Gaussian Noise                | 0.1      | 0.5    |
> > | 37.2        | 0.87   | 1000  | 2      | Gaussian Noise                | 0.1      | 0.05   |
> > | 39.1        | 0.91   | 1000  | 2      | Gaussian Noise                | 0.5      | 0.1    |
> > | 40.2        | 0.94   | 1000  | 2      | Gaussian Noise                | 0.05     | 0.1    |
> > | **42.3**    | **0.96** | 1000 | 2      | Gaussian Noise                | 0.1      | 0.1    |
> > | 34.1        | 0.85   | 500   | 2      | Gaussian Noise                | 0.1      | 0.1    |
> >
> > ### Is the Restoration Operator applied at every diffusion step or intermittently?
> >
> > Yes, both the DIP model and the restoration operator are applied to each reverse step of the diffusion model. However, we run only two DIP epochs per DM step.
> >
> > Finally, if the document is accepted, the corresponding adjustments will be made to address the weaknesses and questions raised.

---

### Official Review · Reviewer_xEuF · 2025-11-02

**Soundness:** 3
**Presentation:** 3
**Contribution:** 3
**Rating:** 4
**Confidence:** 4

**Summary:**

This paper proposes DM-RODIP, which uses the reverse process of a pre-trained diffusion model to generate or reconstruct seismic data. At each step, it uses a recovery operator based on DIP to guide inverse sampling. This approach aims to preserve the statistical prior captured by the diffusion model during the generation process, while also reconstructing the seismic data. Finally, the experimental results using synthetic and measured (Stratton 3D) data demonstrated that DM-RODIP achieves superior reconstruction quality compared to several baseline methods, such as DIP and CCSeis-DDPM.

**Strengths:**

1. Combining pre-trained DM (e.g. capturing global statistics) with DIP (e.g. leveraging structural priors while only requiring observations) is an interesting idea.

2. The proposed solution avoids the need to retrain the entire diffusion model for each specific reconstruction scenario, thereby reducing computational costs.

3. The experimental results on seismic imaging are good, so it might be useful for scenarios involving sparse or mismatched training data, compensating for the limitations of pre-trained diffusion models in domain-specific features.

**Weaknesses:**

Although the solution is interesting, the main idea is not entirely novel, but rather a combination of existing ideas and their engineering implementation, i.e., the method is an extension of CDDIP (Goyes-Penafiel et al., 2025) which adopts an additional existing calibration method (restoration prior).

Although the paper aims to reduce computational costs by eliminating the need to retrain the deep model, the algorithm itself often carries out DIP (with optimisation) and proxy steps during inference, which can be computationally intensive. The paper lacks detailed comparisons of runtime, memory consumption, per-DIP optimisation costs and overall inference time. This makes it impossible to verify the claim that computational costs are lower.  For example, if DIP uses the current $X_{t−1}$ as input for fitting observations at each step, this amounts to iterative optimisation (and calling the forward model multiple times) during the inference phase. Furthermore, to my knowledge, DIP itself suffers from overfitting due to its reliance on early stopping strategies. Solving a DIP subproblem at each step during backpropagation (as per line 4 of Algorithm 1) could be computationally intensive, potentially introducing noise or instability during optimisation. A further clarification is expected.

The comparison methods primarily involve DIP (isolated) and CCSeis-DDPM. Although the paper cites several works on diffusion-guided or projection-related approaches (such as DPS and DiffPIR), it neither quantifies these as strong baselines for comparison nor integrates them.

**Questions:**

Line 288. In Algorithm1, what is the $\sqrt{\Sigma}$?

It's better to present some failed cases and analyse limitations (such as degradation under high noise or extremely sparse sampling conditions).

How to prevent overfitting? As the DIP is involved, this approach risks overfitting observational noise or implicitly reusing observations multiple times.

How do you crop/normalise Straton 3D data?

---

> ### Author Response · Authors · 2025-11-27
>
> Thank you for your feedback. To address the aforementioned weaknesses, we added the execution-time column and the DPS results to Table 3. Note that for all diffusion-based methods, the diffusion model required 35 seconds, and this time was added to the execution time of each solver. We report it this way to isolate and compare the computational cost attributable specifically to each solver.
>
> **Table 3.** Performance assessment on Land Acquisition and Stratton 3D datasets for the DM-RODIP, CCSeis-DDPM, DPS, and DIP methods using PSNR and SSIM metrics.
>
> | Method | Dataset | PSNR (dB) ↑ | SSIM ↑ | Time (s) ↓ |
> |---|---|---|---|---|
> | DM-RODIP | Land Acquisition | **42.3 ± 0.7** | **0.95 ± 0.01** | 1543 (35 + 1508) |
> |  | Stratton 3D | **31.1 ± 0.8** | **0.79 ± 0.04** | |
> | CCSeis-DDPM | Land Acquisition | 32.1 ± 0.07 | 0.86 ± 0.01 | 364 (35 + 329) |
> |  | Stratton 3D | 30.1 ± 0.35 | 0.75 ± 0.01 | |
> | DPS | Land Acquisition | 20.2 ± 0.02 | 0.29 ± 0.01 | 69 (35 + 34) |
> |  | Stratton 3D | 29.3 ± 0.01 | 0.78 ± 0.01 | |
> | DIP | Land Acquisition | 24.6 ± 2.3 | 0.53 ± 0.11 | **125** |
> |  | Stratton 3D | 25.1 ± 1.7 | 0.14 ± 0.07 | |
>
>
> As shown, the proposed method outperforms the DPS method. This is expected as DPS is designed for restoration tasks in natural images. In our scenario, the problem corresponds to an inpainting setting in which entire columns of the image are lost. Consequently, methods that are not focused on this type of degradation exhibit limited performance in reconstructing seismic traces.
>
> In the time column, the proposed method has the longest execution time. However, as noted in the document, its advantage in computational cost lies in the fact that it does not require retraining. In contrast, state-of-the-art methods based on diffusion models must be retrained for each experiment using dense and uniform acquisitions, thereby using part of the available data for that specific acquisition. This retraining overhead constitutes the main computational expense that our method avoids.

---

> ### Author Response · Authors · 2025-11-27
>
> Regarding your questions, we would like to comment on the following:
>
> ### Line 288. In Algorithm1, what is the $\sqrt{\Sigma}$?
>
> Thank you for pointing this out. It is indeed a typo, which has now been corrected in the document; in this case, the correct expression is $\sqrt{\beta_t}$.
>
> ### It's better to present some failed cases and analyse limitations (such as degradation under high noise or extremely sparse sampling conditions).
>
> Thank you for the recommendation. We evaluated the proposed algorithm in scenarios with a higher subsampling rate, considering 70% and 80%, which are presented in Table 4.
>
> **Table 4.** PSNR and SSIM metrics for different subsampling rates in the synthetic Land Acquisition data reconstruction scenario.
>
> | Subsampling Rate | PSNR (dB) ↑ | SSIM ↑ |
> |---|---|---|
> | 20% | 42.4 | 0.97 |
> | 30% | 42.4 | 0.97 |
> | 40% | 42.3 | 0.96 |
> | 50% | 42.3 | 0.95 |
> | 60% | 39.1 | 0.90 |
> | 70% | 37.1 | 0.85 |
> | 80% | 32.9 | 0.79 |
>
> As expected, at higher subsampling rates, performance in the reconstruction task continues to decrease; however, acceptable performance is still maintained for these scenarios of 70% and 80% subsampling rates.
>
> ### How to prevent overfitting? As the DIP is involved, this approach risks overfitting observational noise or implicitly reusing observations multiple times.
>
> We update the DIP at every reverse step (line 4), this inner optimization is intentionally lightweight and decoupled from the diffusion chain: the diffusion model remains fixed and is treated as a constant input to the DIP loss (i.e., no gradients flow through the reverse update in Eq.(7)). At each DM step we run only two DIP epochs with a small learning rate (Sec. 5.2.2), which effectively acts as an implicit early‑stopping mechanism rather than a full DIP training loop. The DIP output influences the iteration solely through the bounded anchor $\mathbf{z}_{t-1}$ and the subsequent proximal data-consistency map in Eq.(11), resulting in a stable quadratic correction.
>
> In our formulation, the "forward model" inside the DIP loss corresponds to the subsampling acquisition model (Eqs.(5)–(6)), so each DIP evaluation reduces to a masked MSE, and the proximal step is closed-form. Classic DIP overfitting concerns are mitigated because the network is never optimized to convergence at any step, the loss is restricted to measured entries via $\mathbf{H}$ (Eq.(9)), unmeasured entries are jointly regularized by the diffusion prior and the proximal projection, and the re‑noising in Eq.(12) injects stochasticity that discourages memorization of acquisition noise.
>
> Empirically, this design yields stable optimization and low variance across realizations (Table 3), while providing the intended guidance effect without retraining the diffusion prior.
>
> ### How do you crop/normalise Straton 3D data?
>
> All data were resized to 128x128, and the Straton 3D data were normalized to the range -1 to 1 using feature scaling.
>
> ---
>
> Finally, if the document is accepted, the corresponding adjustments will be made to address the weaknesses and questions raised, including adjustments to the graphs to add the new results.

---

### Meta-Review · Area_Chair_FTai · 2025-12-30

**Summary:**

his paper proposes a seismic shot-gather reconstruction method that combines a diffusion model prior with Deep Image Prior. Reviewers acknowledged the strong empirical performance of the proposed approach, while raising concerns about its limited novelty and positioning relative to existing work.

In particular, reviewers noted that the experimental comparisons are largely restricted to DIP-based and DDPM-based baselines, while stronger diffusion-guided or plug-and-play approaches are missing. In this context, the paper’s claim that most existing diffusion-based inverse problem methods require retraining appears to be overly broad. Many diffusion-based approaches operate through inference-time guidance or data-consistency mechanisms and do not require retraining, making them directly relevant baselines for comparison.

More generally, the proposed method primarily integrates existing components in a task-specific manner, and its conceptual distinction from prior diffusion-guided reconstruction frameworks is not sufficiently clarified. As a result, the paper lacks a comprehensive discussion and comparison that would be necessary to clearly establish its novelty and positioning in the literature, particularly in terms of reconstruction quality and computational cost.

While the authors added experimental results for a diffusion-guided baseline during the rebuttal, this addition addresses only a single instance within a broader class of relevant methods highlighted by the reviewers. Consequently, the novelty and baseline coverage concerns are not fully resolved.

Overall, I do not believe the paper meets the novelty bar for ICLR in its current form, and I therefore recommend rejection.

**Reviewer Concerns:**

Two of the three reviewers raised concerns regarding the limited baseline comparisons, noting that the evaluation is largely restricted to DIP-based and DDPM-based methods, while stronger diffusion-guided or plug-and-play approaches are missing. In response, the authors added experimental results for DPS during the rebuttal. While this addition is a useful step, it represents only a single instance within the broader class of methods highlighted by the reviewers, and the baseline coverage concerns are therefore not fully resolved.

Two reviewers also expressed concerns about the limited novelty of the work, stating that the proposed method appears to primarily combine existing ideas through an engineering-oriented implementation. This concern was not directly addressed in the rebuttal.

The authors adequately addressed the technical questions raised by the reviewers during the rebuttal.

**Reviewer Scores:**

Reviewer xEuF (score: 4) raised concerns regarding limited novelty, insufficient baseline comparisons, and high computational cost. These concerns largely remain unresolved after the rebuttal, and the score is therefore unlikely to change.

Reviewer xbj5 (score: 6) raised several technical questions, which the authors adequately addressed during the rebuttal. However, the concern regarding insufficient baseline comparisons was only partially resolved, as discussed above. Given the initial score, a further increase is unlikely.

Reviewer Azzx (score: 2) based the low score primarily on concerns about the limited novelty and overall contribution of the paper, as well as the limited experimental evaluation. As these issues were not fully addressed during the rebuttal, a significant increase in the score is not expected.

---

### Decision · Program_Chairs · 2026-01-26

Reject